# Serotonin Transporter (SERT) Expression Modulates the Composition of the Western-Diet-Induced Microbiota in Aged Female Mice

**DOI:** 10.3390/nu15133048

**Published:** 2023-07-06

**Authors:** Mirjam Bloemendaal, Ekaterina Veniaminova, Daniel C. Anthony, Anna Gorlova, Priscilla Vlaming, Adel Khairetdinova, Raymond Cespuglio, Klaus Peter Lesch, Alejandro Arias Vasquez, Tatyana Strekalova

**Affiliations:** 1Departments of Psychiatry & Human Genetics, Radboud University Medical Center, 6525 GA Nijmegen, The Netherlands; priscilla.vlaming@radboudumc.nl (P.V.); alejandro.ariasvasquez@radboudumc.nl (A.A.V.); 2Laboratory of Psychiatric Neurobiology, Institute of Molecular Medicine and Department of Normal Physiology, Sechenov First Moscow State Medical University, 119991 Moscow, Russia; katya.veniaminova@gmail.com (E.V.); anna.gorlova204@gmail.com (A.G.); khairetdinova.adele@gmail.com (A.K.); raymond.cespuglio@univ-lyon1.fr (R.C.); 3Department of Pharmacology, Oxford University, Oxford OX1 3QT, UK; daniel.anthony@pharm.ox.ac.uk; 4Neuroscience Research Center of Lyon, Claude-Bernard Lyon-1 University, 69500 Bron, France; 5Division of Molecular Psychiatry, Center of Mental Health, University of Würzburg, 97080 Würzburg, Germany; kplesch@mail.uni-wuerzburg.de (K.P.L.); t.strekalova@maastrichtuniversity.nl (T.S.); 6Department of Psychiatry and Neuropsychology, School for Mental Health and Neuroscience, Maastricht University, 6229 HX Maastricht, The Netherlands

**Keywords:** gut microbiome, SERT, serotonin, Western diet, metabolic health, brain, behavior

## Abstract

**Background.** The serotonin transporter (SERT), highly expressed in the gut and brain, is implicated in metabolic processes. A genetic variant of the upstream regulatory region of the *SLC6A4* gene encoding SERT, the so-called short (s) allele, in comparison with the long (l) allele, results in the decreased function of this transporter, altered serotonergic regulation, an increased risk of psychiatric pathology and type-2 diabetes and obesity, especially in older women. Aged female mice with the complete (*Sert*^−/−^: KO) or partial (*Sert*^+/−^: HET) loss of SERT exhibit more pronounced negative effects following their exposure to a Western diet in comparison to wild-type (*Sert*^+/+^: WT) animals. **Aims.** We hypothesized that these effects might be mediated by an altered gut microbiota, which has been shown to influence serotonin metabolism. We performed V4 16S rRNA sequencing of the gut microbiota in 12-month-old WT, KO and HET female mice that were housed on a control or Western diet for three weeks. **Results.** The relative abundance of 11 genera was increased, and the abundance of 6 genera was decreased in the Western-diet-housed mice compared to the controls. There were correlations between the abundance of *Streptococcus* and *Ruminococcaceae_UCG-014* and the expression of the pro-inflammatory marker Toll-like-Receptor 4 (*Tlr4*) in the dorsal raphe, as well as the expression of the mitochondrial activity marker perixome-proliferator-activated-receptor-cofactor-1b (*Ppargc1b*) in the prefrontal cortex. Although there was no significant impact of genotype on the microbiota in animals fed with the Control diet, there were significant interactions between diet and genotype. Following FDR correction, the Western diet increased the relative abundance of *Intestinimonas* and *Atopostipes* in the KO animals, which was not observed in the other groups. *Erysipelatoclostridium* abundance was increased by the Western diet in the WT group but not in HET or KO animals. **Conclusions.** The enhanced effects of a challenge with a Western diet in SERT-deficient mice include the altered representation of several gut genera, such as *Intestinimonas*, *Atopostipes* and *Erysipelatoclostridium*, which are also implicated in serotonergic and lipid metabolism. The manipulation of these genera may prove useful in individuals with the short SERT allele.

## 1. Introduction

A genetic variant of the (Solute Carrier Family 6 Member 4) *SLC6A4* gene, the so-called short allele, results in the decreased function of the serotonin transporter (SERT) in comparison with the long allele, as well as in a stress-related vulnerability to anxiety and depression [1,2,3]. The short allele is also associated with an increased risk of type-2 diabetes [4,5] and obesity [6,7], which is more pronounced in older females [8,9,10]. Today, more than one in two adults, and nearly one in six children, are overweight or obese in the OECD area, i.e., first-world countries [11], which is exacerbated by the increased intake of the so-called “Western diet”, which is a hypercaloric diet enriched with saturated fat, cholesterol and sugars [12].

Carrying the short allele results in both central and peripheral dysregulation of serotonin re-uptake and metabolism [13], as well as a spectrum of pleiotropic physiological consequences following the excessive consumption of a high fat/hypercaloric diet [14,15,16]. The mechanisms of this gene × environment interplay remain elusive. In addition, aging is known to suppress SERT expression and activity [17,18,19,20], which are implicated in mitochondrial dysfunction and compromised tissue energy metabolism [21,22,23].

The gut contains 10^14^ microorganisms that may play a pivotal role in the interplay between SERT expression and the consumption of a Western diet, the latter of which is the biggest driver of changes in the composition of the gut microbiota [24]. Recently, certain changes in intestinal microbial composition have been reported to contribute to decreased glucose tolerance, insulin resistance and obesity. For example, germ-free mice (mice born and raised in a sterile environment, preventing the colonization of the gut with microbes) have been shown to remain lean on a Western diet [25]. The abundance of the gut bacterium *Parabacteroides distasonis,* one of the bacteria involved in transforming bile acids [26,27,28], has been shown to be lower in obese individuals, and supplementation with *P. distasonis* in mice counteracts obesity and the hyperglycemia that is induced by a high-fat diet [29]. Exposure to a high-fat diet has also been shown to enhance susceptibility to intestinal inflammation that is associated with an increased abundance of pro-inflammatory bacteria, such as *Escherichia coli* and the mucus degrading bacterium *Akkermansia muciniphilia* [30,31,32].

Indeed, reduced SERT activity [33], metabolic dysfunction associated with excessive intake of a Western diet [30,31,32] and aging [34,35] have all been associated with specific changes in the gut microbiota, but little has been done to study the interactions with gut microbiota and their contribution to metabolic abnormalities.

It is noteworthy that over 90% of the serotonin in the body is produced in the gut, where it is involved in the regulation of peristalsis, the maintenance of enteric neurons and lipid absorption [33]. Disturbances in serotonin metabolism and the gut microbiota have been argued to mediate the physiological effects of the Western diet [33,36,37,38]. For example, housing rats on a high-fat diet increases levels of intestinal serotonin, deoxycholic acid and cholic acid, the regulators of lipid and bile acid metabolism, and accelerates gastrointestinal transit, all of which are frequently associated with obesity. These abnormalities are reversed when rats are given a microbiota transplant from rats fed a normal diet [38]. Host SERT expression can also affect gut serotonin levels, and gut serotonin was found to be elevated in mice lacking SERT [33,39]. The genetic lack of SERT was discovered to decrease the abundance of Akkermansia, a bacterium that is known to be affected by the Western diet, and it elevates the abundance of *Lactobacillus* sp. A genetic lack of SERT can also lead to the decreased abundance of *Bifidobacteria*, which regulates the gut’s functionally important synthesis of short-chain fatty acids, and it increases abundance in spore-forming bacteria (such as *Turicibacter sanguinis*), promoting serotonin production by enterochromaffin cells and regulating triglyceride metabolism and adipose tissue [40,41].

Thus, the available literature illustrates that there are profound effects of the Western diet and the level SERT expression on gut microbiota, and an interaction is likely. However, this interplay has not been systematically addressed in mechanistic studies, and most experiments have been conducted on young male mice, which are least susceptible to metabolic aberrations associated with alimentary conditions and SERT functions, as compared to aged females. In a previous study, 12-month-old female mice with the complete (*Sert^−/^*^−^: KO) or partial (*Sert^+/−^*: HET) loss of SERT were found to have overly elevated metabolic, molecular and behavioral responses to a 3-week-long period of exposure to the Western diet, when compared with wild-type (*Sert^+/+^*: WT) animals [42]. Interactions were observed for obesity, glucose tolerance, the expression of Toll-like receptor 4 (*Tlr4*) in the brain, the expression of perixome-proliferator-activated-receptor-cofactor-1b (*ppargc1b*) in the brain and liver and hippocampus-dependent performance and emotionality. However, not all changes in gene expression in the brain and liver of KO mice were exhibited by the HET mice fed with the Western diet, and all three possible genotypes exhibited distinct Western-diet-induced changes. In the present study, we sought to expand this work by analyzing the effect on the gut microbiome of a 3-week-long exposure to the Western diet in 12-month-old WT, KO and HET female mice using V4 16S rRNA sequencing, and these changes were related to the previously reported brain expression of *Tlr4* and *ppargc1b* in specific regions within the brain and in the liver [42].

## 2. Methods

### 2.1. Animals

Experiments were carried out on 12-month-old SERT KO, HET and WT littermate female mice; breeding was carried out as described elsewhere [42]. Mice were housed in groups of five under standard laboratory conditions (22 ± 1 °C, 55% humidity, food and water ad libitum) and a reversed 12 h light–dark cycle (lights on: 21:00 h, lights off: 09:00 h, to accommodate their natural circadian rhythm). All experimental conditions were set up and maintained in accordance with the European Communities Council Directive for the care and use of laboratory animals (2010/63/EU) and the ARRIVE guidelines (http://www.nc3rs.org.uk/arrive-guidelines (accessed 30 June 2023)), approved by the local ethics committee of Oxford University.

### 2.2. Study Design and Dietary Intervention

Mice of all three genotypes were assigned to groups of 6–8 animals that received either the control or “Western diet” feeding regimes (D18071801, energy content of 3.8 kcal/g, 4.3% total fat, 1.3% saturated fat or D11012302, energy content of 4.6 kcal/g, 0.2% cholesterol, 21.3% fat, 10.5% saturated fat, respectively; Research Diet Inc., New Brunswick, NJ, USA) for 21 days, as described elsewhere ([42,43,44,45]; Figure 1A,B). For further details on the diets, see the Appendix A. On day 0 (‘Pre-intervention’, time point 1), fecal material from all groups of mice was collected by placing them on an open field arena. Mice were weighed on day 1 and day 21 of the dietary intervention, and body weight changes were calculated. On day 21, mice were studied for glucose tolerance and behavior (data reported in [42]). On day 23 (‘Post-intervention’, time point 2), mice were terminally anesthetized with isoflurane inhalation, their visceral abdominal fat was dissected and weighed, and fecal samples were collected. The liver, hippocampus, prefrontal cortex hypothalamus and dorsal raphe were isolated for further RNA isolation and PCR of *Tlr4* and *Ppargc1b* (data are reported in [42]), with sterile forceps. All material was stored at −80 °C until use.

### 2.3. Bacterial DNA Isolation and Sequencing

V4 16S rRNA sequencing of a total of 88 fecal samples was performed (see Appendix A). Microbial DNA was isolated and purified using the bead-beating procedure protocol of Baseclear B.V. (Leiden, The Netherlands). The amplification of the V4 region of the 16S ribosomal RNA (rRNA) gene was run in duplicate PCR for each sample, and the 515F primer (5′GTGYCAGCMGCCGCGGTAA) and 806R (5′GGACTACNVGGGTWTCTAAT) were used. In order to rule out possible confounders, one PCR negative control sample and two positive control samples containing known microbial compositions were included. Libraries were pooled for sequencing on the Illumina NovaSeq 6000 platform (paired-end, 250 bp) (Baseclear B.V., Leiden, The Netherlands). Sequence reads were demultiplexed and filtered using Illumina Chastity, and reads containing the PhiX control signal and adapter sequences were removed from all reads. An average sequencing quality score “Q“ of 36.1 (range 35.51–36.29) was obtained, indicating the sequencing error probability for a given base, where Q = 30 indicates 99.9% accuracy. We observed a total number of raw read pairs of 74,544,944 with an average of 847,192 and a range of 119,145–1,242,224.

### 2.4. Bioinformatics

Reads were processed for statistical analysis using QIIME2 [46]. First, reads were denoised using DADA2 [47]: primers were trimmed, sequencing errors (chimera’s) were removed, read pairs were combined, and erroneous read combinations were removed. Read pairs were aligned into Amplicon Sequence Variants (ASVs) using mafft [48]. Combined reads deviating from the expected combined length of 250 bases were filtered out, allowing variations of 10 bases in both directions (i.e., 240–260 range), which was 0.1% of the total of 56,023,158 reads. This resulted in a total of 55,969,125 reads corresponding to 2823 unique ASVs over the 88 samples. Taxonomic assignment was performed in the q2-feature-classifier [49] using the Naive Bayes classifier trained on the SILVA reference database (version 132.99% OTU from 515/806R region of sequences) [50].

A phylogenetic tree was built using fasttree2 [51] and combined in a biome file including ASV and taxonomy information. Non-bacterial and unassigned ASVs were removed from the biome file. An average read count per sample of 668,794 was obtained, ranging between 100,761 and 974,171 (see Appendix A). These sequences represented a total of 2623 ASVs with an average of 393 ASVs (per sample), ranging between 167 and 568 ASVs. Data were made compositional, i.e., the abundance of each ASV was expressed relative to the total number and prevalence of ASVs.

### 2.5. Statistical Analysis

The outcome measures in mice were analyzed using GraphPad Prism version 8.01 (San Diego, CA, USA). The normality of the data distribution was tested using the Shapiro–Wilk normality test; as all the data were distributed normally, a two-way ANOVA followed by Tukey’s multiple comparisons test was used. The level of significance was set at *p* < 0.05. Data are presented as mean ± SEM.

### 2.6. Community Analyses of Microbiota

All statistical analyses were performed in R (version 3.6.2) using the software package phyloseq [52] and microbiome [53], and other packages are cited where they are used. Global community analyses were performed without any prevalence filtering at the genus and sample level to prevent inducing a bias against less prevalent genera. Community analyses are reported in the Appendix A.

### 2.7. Compositional Analyses on the Taxonomic Data

Compositional analyses were conducted at the genus level. Two filtering steps were applied to reduce excess zeros in the composition data (see [54]). First, the 2623 ASV’s were aggregated into 345 genera. Of these, 208 genera with a prevalence of <10% across all samples were removed, resulting in 137 remaining genera. Second, we studied if any samples showed less than 10% of the genera with non-zero values, and no such cases were found.

For each genus, the relative abundance values of the post-dietary intervention were corrected for their own pre-intervention values by calculating the change in the relative abundance over time, referred to as the delta values, symbolized by ‘Δ’. Non-parametric regressions were performed on the delta values of these 137 genera using the raov function of the Rfit package [55], testing for the main effects of diet and diet × genotype interactions. In short, this entails least-squares estimation on the ranks of the residual scores rather than the raw data to account for the skewed nature of the data. Rank-based statistics are very commonly used in non-parametric testing, as the analysis is less sensitive to extreme values in the distribution [55]. The effects of genotype were tested on the abundance values of the control group on day 0, referred as time point 1 (hereby excluding the effects of the dietary intervention), with the WT group as a reference.

In the case of FDR (false discovery rate)-corrected significant main or interaction effects, simple effects were tested using the Wilcoxson signed-rank test; for the main effects of diet, no further tests were performed, as this main effect reflects a difference between two groups: the Western diet group and the Control diet group.

The visual data inspection of the relative abundance distribution included listing the number of non-zero and zero observations per group. As it is impossible to dissociate between a true absence and a presence below the detection limit of the sequencing technique, results based on very low non-zero numbers should be treated with caution. Last, extreme delta values are determined by delta values either above Q3 + 3xIQR or below Q1 − 3xIQR. For the assessment of significant effects per genus, these extreme values were excluded in order to assess the robustness of the effect.

### 2.8. Exploratory Association Analyses between Gut Microbiota and Behavioral, Metabolic and Brain Measures

For those 19 genera that were altered by diet, we performed Spearman’s correlations with several relevant physiological parameters in dietary challenged mice. These parameters were selected based on the fact that they were significantly affected by diet and included body weight, body fat, *Tlr4*mRNA concentrations in the dorsal raphe and *Ppargc1b*mRNA expression in the dorsal raphe, prefrontal cortex and liver (reported in [42]). Significant correlations were studied for extreme values (i.e., values 1.5 times the interquartile range outside the third or first quartile). Correlations driven by such values were not taken into consideration and are not reported. In the case of a significant correlation in the Western diet group not driven by outliers, the correlation within the Control diet group was calculated to find out whether this effect was selectively present in the Western diet group, and correlations in both groups were plotted. As these are exploratory analyses, no multiple correction was applied.

## 3. Results

### 3.1. Physiological Changes in Host Metabolism and Brain Gene Expression

A significant diet effect (F = 67.53, *p* < 0.0001, two-way ANOVA), but not a genotype effect (F = 0.292, *p* = 0.748) or an interaction (F = 1.805, *p* = 0.178), was present for body weight gain between D21 and D1. Body weight gain was higher in the groups fed with the Western diet (*p* < 0.0001 for all the genotypes, Tukey’s test; Figure 1D). The visceral fat percentage of the total body weight was significantly affected both by diet (F = 10.01, *p* = 0.0031) and genotype (F = 3.786, *p* = 0.0317), but not their interaction (F = 2.705, *p* = 0.0797). A post hoc group comparison revealed a significant increase in visceral fat percentage in Western-diet-fed KO mice compared to the KO group fed with the Control diet (*p* = 0.0141, Tukey’s test; Figure 1E). The effects of dietary intervention on *Tlr4* mRNA concentrations in the dorsal raphe and *Ppargc1b* mRNA expression were reported by [42]. Briefly, the principal changes induced by the Western diet were a significant increase in *Tlr4* mRNA concentration in the dorsal raphe and decreases in *Ppargc1b* mRNA levels in the dorsal raphe, prefrontal cortex and liver.

### 3.2. Community Analyses of Gut Microbiota

For alpha diversity, we observed a main effect of time, but no effect of diet or genotype. For beta diversity, an interaction between time*diet and time*genotype was observed. For more information, see the Supplementary Results in Appendix A.

### 3.3. Compositional Analyses of Gut Microbiota

In the 20 genera, we observed an FDR-corrected significant effect of either diet and/or an interaction between these measures for relative abundance per group (Table 1, Figure 2 and Figure 3, Appendix A). A main effect of diet was shown for 17 genera, and an interaction between diet and genotype was found in three genera. Within each genus that was significantly affected, we calculated whether this significance was due to extreme Δ values. Extreme values were defined as those above Q3 + 3xIQR or below Q1 − 3xIQR in the diet and/or genotype groups and were omitted from the analysis.

### 3.4. Effects of Diet on Microbiota

In 17 genera, a main effect of diet was observed (Figure 2). Six of these genera belong to the *Ruminococcaceae* family, three genera belong to the *Erysipelotrichaceae* family, and a single genus per family was found for *Peptococcaceae*, *Streptococcaceae*, *Staphylococcaceae*, *Rikenellaceae*, *Desulfovibrionaceae*, *Peptostreptococcaceae*, *Lachnospiraceae* and *Staphylococcaceae*. After a 21-day period of dietary intervention, challenged versus control groups showed an increase in relative abundance of eleven genera: *Ruminiclostridium.9*, *Oscillibacter*, *Ruminococcaceae.uncultured*, *Ruminiclostridium.5*, *Eubacterium.coprostanoligenes.group*, *Lachnospiraceae(f).A2*, *Peptococcaceae(f).uncultured*, *Streptococcus*, *Staphylococcus*, *Alistipes* and *Bilophila*, and a decrease was observed in *Faecalibaculum*, *Romboutsia*, *Mollicutes(o).RF39(f). uncultured*, *Turicibacter*, *Dubosiella* and *Ruminococcaceae.UCG.014*. 

In three genera, significant changes were due to large numbers of zero observations. For *Romboutsia*, this result was based on zero observations at time point 1 (in all genotype groups); at time point 2, both the control and Western diet groups had all or ≥4 non-zero observations. For the genus *Mollicutes* at time point 2, the Western diet KO group had four non-zero observations. For the genus *Turicibacter*, there were ≤4 non-zero observations at time point 1 in the WT and KO Control diet groups and in the KO group housed on the Western diet.

### 3.5. Effects of Genotype

No FDR-significant effects of genotype were observed (all corrected *p*-values > 0.05).

### 3.6. Diet × Genotype Interactions

For the genus *Atopostipes* (*Carnobacteriaceae family*), *Intestinimonas* (*Ruminococcaceae family*) and *Erysipelatoclostridium* (*Erysipelatotrichaceae family*), a diet × genotype interaction was revealed (Figure 3). The relative abundance of *Atopostipes* and *Intestinimonas* was higher in KO mice challenged with the Western diet in comparison with other groups (Mann–Whitney U test was used to compare the former group with the KO group housed on the Control diet and with other groups; all tests *p* < 0.009). Notably, at time point 2, the genus Atopostipes only had >1 non-zero observations in one group for KO mice fed with the Western diet (see also Appendix A). The other groups consist of zero or one non-zero observation; hence, these results should be interpreted with caution. At time point 2, the genus Erysipelatoclostridium exhibited an increased abundance in the WT group that was housed on the Western diet. The relative abundance in this group was significantly different from the HET control group (*p* = 0.03), but not from other groups (*p* values > 0.1). At time point 2, some groups in this genus had zero or one non-zero observation, such as the WT and HET mice that received the Control diet and KO mutants that were fed with the Western diet (Appendix A). 

FDR-corrected significant main effects of diet and genotype were observed in the genera *Atopostipes* and *Intestinimonas*, and for *Erysipelatoclostridium*, the main effect of genotype was found (Table 1). The interaction term indicates that the main effects of diet and genotype are conditional on the other factor, e.g., a main effect of diet is driven by one genotype group. Hence, these main effects were not explicitly considered for the genera in which an interaction was observed.

### 3.7. Associations between the Effects of the Western Diet on the Gut Microbiota and Cellular and Molecular Changes

There was a significant positive correlation between the visceral fat and abundance of the genus *Coprostanoligenes* (Rho Western-diet-fed group = 0.47, *p*-value Western diet group = 0.03, Rho Control Diet group = 0.3, *p*-value Control Diet group = 0.15, Figure 4A), i.e., the increased abundance of the genus *Coprostanoligenes* in the Western-diet-fed group was associated with increased visceral fat. There was a correlation between *Tlr4* mRNA concentration in the dorsal raphe and the abundance of *Ruminococcaceae_UCG-014* (Rho Western-diet-fed group = −0.48, *p*-value Western diet group = 0.03, Rho Control diet group = −0.1, *p*-value Control diet group = 0.7, Figure 4B). We also found a negative correlation between *pparg1b mRNA* concentrations in the prefrontal cortex and the abundance of *Streptococcus* in the Western-diet-fed group (Rho Western diet group = 0.48, *p*-value Western diet group = 0.03, Rho Control diet group = 0.3, *p*-value Control diet group = 0.9, see Figure 4C). No other correlations were observed for any of the other 19 genera associated with weight changes in the Western diet and *pparg1bmRNA* concentrations in the dorsal raphe.

## 4. Discussion

### 4.1. Summary

We found effects of the Western diet on the gut microbiota both at the community level as well as the taxonomic level. The Western diet differentially affected beta diversity compared to the Control diet group. Moreover, the abundance of 11 genera increased, and 6 were decreased via consumption of the Western diet, which is likely to be a contributing factor to the negative impact of the Western diet on host health. Indeed, Western-diet-induced changes in the genera *Coprostanoligenes*, *Streptococcus* and *Ruminococcaceae_UCG-014* correlated with important negative physiological consequences of the dietary challenge in mice, such as an increase in a mass of visceral fat, a decrease in mitochondrial function marker *Ppargc1b* mRNA levels in the prefrontal cortex and the expression of *Tlr4* mRNA in the dorsal raphe. Interactions between diet and genotype were present only at the compositional level, where the microbiome of KO mice appeared to be more susceptible to the effects of the Western diet challenge, as characterized by increases in the abundance of the genera *Atopostipes* and *Intestinimonas*, which was not observed in the HET and WT groups. Conversely, the WT challenged group displayed the increased abundance of the genus *Erysipelatoclostridium*, which was not shown for the HET and KO groups fed with the Western diet. Thus, our findings support previous observations that the effects of the Western diet are, in general, exacerbated in aged female mice with genetic *Sert* deficiency; however, all three genotypes, WT, KO and HET mice, displayed distinct changes. This diversity in microbiota changes between the genotypes is consistent with previously reported distinct physiological and molecular responses to the Western diet (glucose tolerance, neuroinflammation, emotional and cognitive behaviors) found in the same cohorts of animals [42]. Thus, differential microbiome responses of the mice of the three genotypes to the Western diet are likely to play a role in mediating the physiological changes that were induced by the dietary challenge.

### 4.2. Effects of the Western Diet on Gut Microbiota

A total of 7 out of the 11 genera whose abundance was increased in the groups of mice fed with the Western diet were previously reported to be increased in adult mice after their exposure to similar dietary regimens on a Western-type or cafeteria-like diet. For example, changes in *Ruminiclostridium.9*, [56,57,58], *Oscillibacter* [59,60,61,62,63], *Ruminococcaceae.uncultured* [64], *Lachnospiraceae(f).A2* [65], *Peptococcaceae(f).uncultured* [27,66], *Alistipes* [67] and *Bilophila* [68] have been reported. We found increases in the abundance of *Staphylococcus*, which has not been associated with the effects of the Western diet to date but has been associated with obesity and serum cholesterol in pregnant women, was lower in lean than in overweight children [69]. Dissimilar to our study, exposure to a high fat diet was previously shown to decrease the abundance of *Ruminiclostridium.5* [31,70], *Eubacterium.coprostanoligenes.group* [71,72] and *Streptococcus* [73]. Of note, the *Ruminiclostridium.5* genus is known only for its associations with inflammatory diseases, such as acute necrotizing pancreatitis [74,75]. Concerning the *Coprostanoligenes* group, this genus is involved in the catabolism of cholesterol [72]; hence, its increase in abundance in the Western-diet-fed group likely owes to the high availability of cholesterol in the gut as an important component of the Western diet. 

Of the six genera with reduced gut abundance in mice exposed to the Western diet, a reduction in the genus *Ruminococcaceae.UCG.014* was also found after the consumption of a high fat diet [72] and in a rat model of necrotizing pancreatitis [76]. There have been some inconsistent results concerning the effects of the Western diet that have been published regarding other genera. For example, the consumption of a Western diet was reported to decrease the abundance of *Faecalibaculum* [66,77], and a positive association between this genera and weight gain has been described [41]. A high fat diet has been shown to increase the abundance of *Romboutsia* in some studies [77] or result in opposite changes in other experiments [78]. The consumption of a high fat diet was previously shown to increase the abundance of *Turicibacter* [79,80] and of *Dubosiella* [78,81,82], which contradicts our experience with these genera.

### 4.3. Exploratory Correlations between Microbiota Changes and Physiological Changes in Mice Challenged with a Western Diet

In mice fed with the Western diet, correlation analyses revealed several associations between microbiota changes and markers of inflammatory and metabolic processes in the brain. Specifically, animals fed with the Western diet exhibited a decreased abundance of *Ruminococcaceae.UCG.014* and also displayed lower gene expression of the inflammatory marker TLR4 in the dorsal raphe. This may suggest a potential role for this genus in suppressing increased pro-inflammatory signaling, which is a common feature in Western-diet-fed animals [43,44]. The dorsal raphe is the largest serotonin-producing brain nucleus [83] and is involved in metabolism and body weight regulation [84]. Interestingly, *Ruminococcaceae.UCG.014* abundance has been found to correlate with fecal serotonin levels in chickens with a systemic infection [85]. Future studies are warranted to address the specific mechanism linking the functions of *Ruminococcaceae.UCG.014*, brain *TLR*4-mediated signaling and the serotonergic regulation of metabolism via the dorsal raphe.

In animals fed with the Western diet, an increase in the abundance of *Streptococcus,* a genus involved in fatty acid biosynthesis [86], was associated with the lower gene expression of the mitochondrial activity marker *Pparg1b*. A link between *Pparg1* expression and the gut microbiota has been demonstrated previously; [25] described the relationship between the effects of microbiota on host energy metabolism and lipid storage, which were putatively mediated via altered *Pparg1* expression. Notably, germ-free mice were reported to be resilient to obesity induced by exposure to the Western diet, which was explained by the altered skeletal expression of a similar receptor, *Pparg1a*. The latter was suggested to be functionally related to the expression of genes encoding enzymes that metabolize fatty acids originating from the gut microbiome [25]. The decreased expression of *Pparg1a* and *Pparg1b* in the prefrontal cortex, the hippocampus, liver and adipose tissue was shown to be related to aging, exposure to highly caloric diets and other factors that slow down the metabolic rate [42,43,87,88]. Present data also suggest that the negative metabolic effect of the Western diet on mitochondrial function in the brain is potentially mediated via the microbiota, specifically *Streptococcus*. Again, further studies are needed to address and investigate specific mechanisms relating the changes in the abundance of *Streptococcus* and the brain expression of mitochondria activity markers, such as *Pparg1*.

### 4.4. Distinct Microbiota Effects of Western Diet on Microbiota of Mice with Differential SERT Gene Expression

Three genera may contribute to the enhanced physiological responses of KO mice exposed to a Western diet. In the Western-diet-challenged mice, the abundance of the genera *Atopostipes* and *Intestinimonas* was similar in the WT and HET groups, whereas KO mice demonstrated an increase in *these* genera. Previously, the abundance of *Atopostipes* was shown to be higher in mice receiving a normal-fat diet than in mice fed with a high-fat diet [63]. The production of branched-chain fatty acids, phenol, indole and tryptophan, among others, by *Atopostipes* [89,90], was found to be elevated in rodents lacking *SERT* [39]. The selective increase in *Atopostipes* in the KO group exposed to the Western diet may owe to the high content of saturated fat in the Western diet, as the increased *Atopostipes* abundance was shown to be associated with hyperlipidemia [91], a well-demonstrated feature of KO mice [42]. However, because the sample with *Atopostipes* counts contained many zero observations, further studies are needed to support our conclusion.

We also found that the KO group exposed to the Western diet increased the high abundance of the genus *Intestinimonas*, which is a butyrate-producing strain and an important regulator of gut motility and other functions relating to gut health [92]. Similar changes in *Intestinimonas* were reported in other studies on mice exposed to a high-fat diet [59,93]. The increase in *Intestinimonas* abundance may be driven by its functional role in the metabolism of tryptophan and serotonin; it is increased in mice with a genetic deficit of *SERT* (known to increase gut serotonin levels) and after tryptophan supplementation (in the offspring of mice with chronic kidney disease) [94]. Beyond the gut, *Intestinimonas* abundance inversely correlates with tryptophan plasma levels in a kidney disease model in rats [95] and positively correlates with serotonin blood levels [58,85]. The abundance of *Intestinimonas* in the gut is suggested to be negatively regulated by aryl hydrocarbon receptors (AhR), which are activated by tryptophan metabolites [96]. Overall, our data suggest that *Intestinimonas* is one of the bacteria that likely mediate the physiological effects of the Western diet under the conditions of a genetic lack of SERT.

Finally, we found that WT mice, but not HET nor KO mice, displayed an increase in the abundance of the genus *Erysipelatoclostridium*, a bacterium whose abundance was also elevated in other studies using high-fat or high-protein diets [97,98,99]. Mice fed with a high fat diet, whose guts were mono-colonized with the *Erysipelatoclostridium* genus, *C.ramosum*, showed a substantial gain in body weight and an increase in body fat percentage [100]. These mice were also shown to display elevated serotonin levels along with elevated expression levels of genes moderating intestinal lipid storage, suggesting a role for the genus *Erysipelatoclostridium* in the regulation of gut serotonin and the mechanisms of dietary-induced obesity. Notably, our study revealed *C. ramosum* as the most frequent species observed among the five species within the genus *Erysipelatoclostridium* (constituting 98.8% of the genus reads) (Appendix A). Together, it can be hypothesized that, due to excessive levels of gut serotonin in mutant mice with SERT deficiency, there was no diet-induced increase in the *Erysipelatoclostridium* genus that was found in the HET and KO animals fed with the Western diet.

### 4.5. Genetic SERT Deficiency and Gut Microbiota

Overall, no robust effect of genotype at either the community or compositional level was revealed in our study. Other groups have reported the reduced gut abundance of *Akkermansia* and the increased abundance of the *Turicibacter* species under the conditions of a complete genetic lack of SERT [37,40]; these microbiome changes were suggested to be relevant in mediating the effects of the Western diet and to be involved in altered lipid metabolism in these mutants. Here, no significant results were found in the abundance of the genus *Akkermansia.* The abundance of the genus *Turicibacter* was not increased differentially between genotypes. A discrepancy from formerly reported results might come from the fact that we used female mice of old age, which is different from previously employed conditions in similar studies, as aging can significantly affect the gut microbiota [35,101].

### 4.6. Effects of Time

We observed main effects for time, i.e., changes in alpha (reduced) and beta diversity over time were also present in the control group. This may have been caused by the progression of the aging process of these 12-month-old mice during the three-week study duration, which is known to affect alpha diversity [35]. Other stress or housing conditions during the experiment may have equally affected the gut microbiota of the aged mice. Another factor of influence may be the shift in the Control diet consumed until arrival at the test location to the one used for the duration of the study, even though the contents of these Control diets were similar. A time* genotype interaction was observed for beta diversity, i.e., time affected the beta diversity of all genotype groups. When considering only time point 1, beta diversity between the HET and KO groups was different, and at time point 2, the WT and KO groups were different (see Appendix A). Again, possibly further aging or housing conditions may have affected genotype groups in different ways [101]. Critically, the effects of diet and their interaction are accounted for by changes over time, i.e., for alpha and beta diversity, a differential effect of diet on the interaction with time was tested. At the taxonomic level, the effects of time were filtered out by using delta values (abundance at week 3 minus the baseline) in the models.

## 5. Conclusions

In aged female mice genetically fully or partially lacking SERT, a 21-day exposure to the Western diet significantly impacted the diversity of the gut microbiome in a manner that was distinct in all three genotypes. A distinct metabolic, neuroinflammatory and behavioral profile is reported for the three genotypes following the challenge with a Western diet. The results described here suggest that there is an inter-relationship between the microbiome changes and physiological effects induced by the Western diet under conditions of genetically determined SERT deficiency. Significant correlations between changes in gut bacteria and the gene expression of key markers of the inflammatory effects of the Western diet in the SERT model, such as *TLR*4 gene over-expression in the dorsal raphe and downregulation of the mitochondrial activity marker Pparg1b in the prefrontal cortex, further support this view. In particular, the enhanced effects of a challenge with a Western diet in SERT-deficient mice include the altered representation of several gut genera, including *Intestinimonas*, *Atopostipes* and *Erysipelatoclostridium*, which are also implicated in serotonergic and lipid metabolism. The manipulation of these genera may prove useful in individuals with the short SERT allele.

## Figures and Tables

**Figure 1 nutrients-15-03048-f001:**
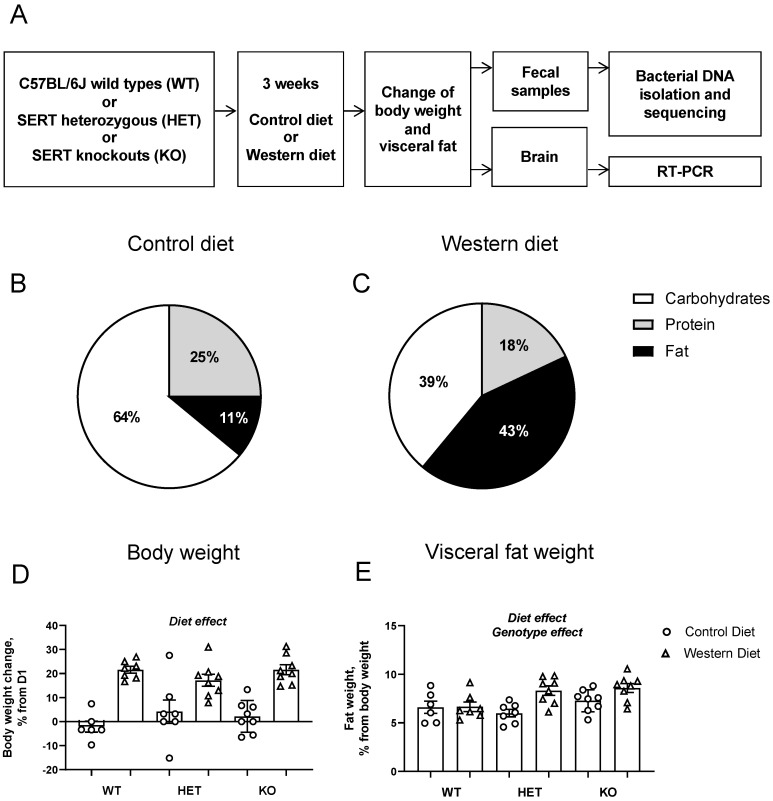
(**A**) Study design. (**B**,**C**) Composition of the Western and Control diet, respectively. (**D**,**E**) Effects of Western diet and SERT deficiency on body weight and visceral body fat, respectively. A main effect of increased body weight and visceral fat was observed as well as a main effect of genotype on visceral fat. Abbreviations: Wild Type (WT), Heterozygotes (HET), Knockout (KO), Serotonin Transporter (SERT), Real-Time Polymerase Chain Reaction (RT-PCR).

**Figure 2 nutrients-15-03048-f002:**
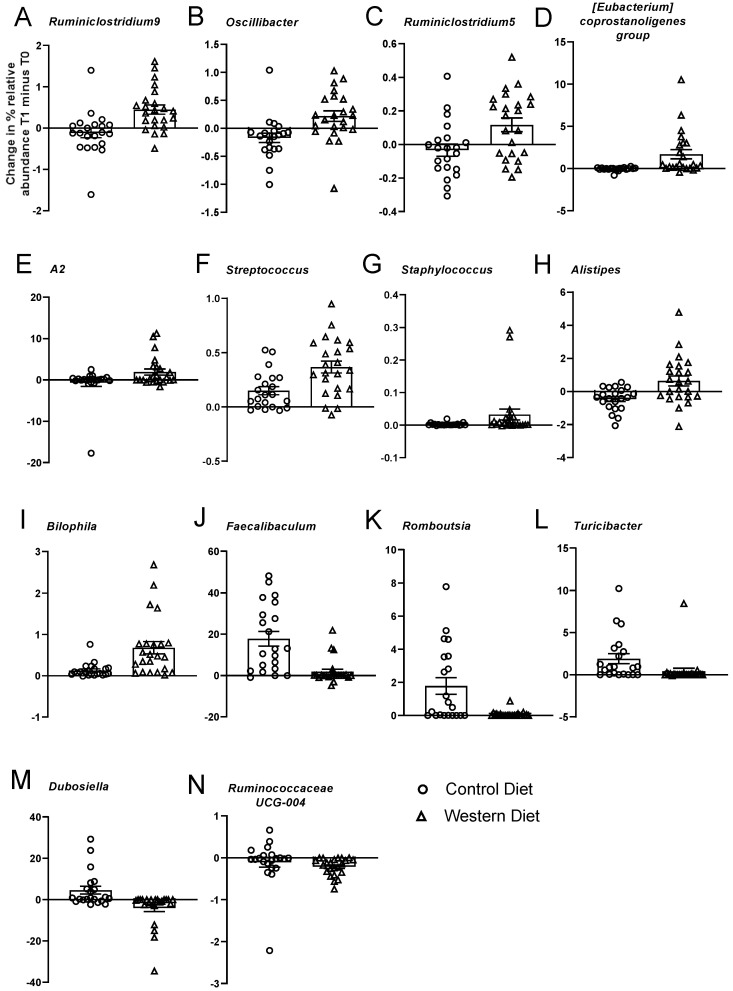
**Significant effects of housing on Western and Control diet on gut microbiota.** Genera of increased abundance are plotted first followed by genera of decreasing abundance. Three uncultured genera are not plotted. Panels (**A**–**I**) show genera with increased abundance after exposure to Western diet, and panels (**J**–**N**) display genera whose abundance was increased after housing on Control diet.

**Figure 3 nutrients-15-03048-f003:**
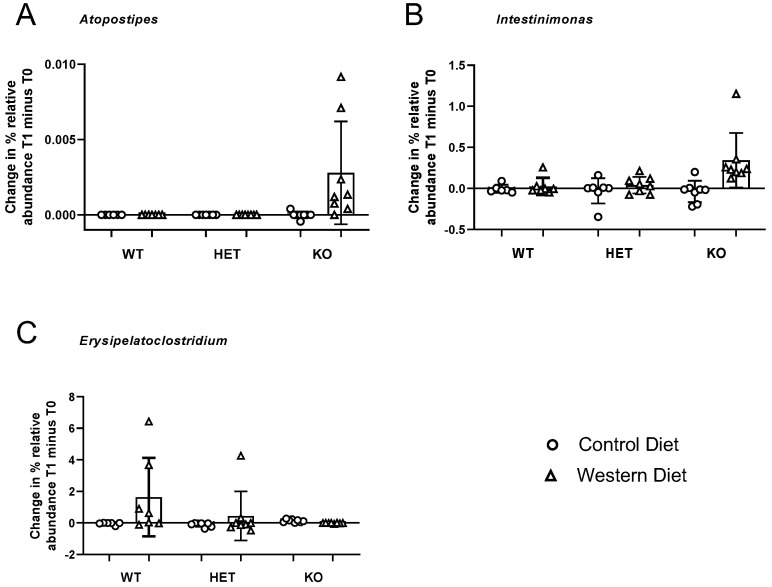
**Significant interaction of gut microbiota and genotype**. Delta values are shown for diet and genotype groups. Genera *Atopostipes* and *Intestinimonas* selectively increased in abundance after Western diet in the KO, but not in the HET and WT groups. Conversely, genus *Erysipelatoclostridium* selectively increased abundance in the WT group, but not in the HET and KO groups. **Abbreviations:** Wild Type (WT), Heterozygotes (HET), Knockout (KO).

**Figure 4 nutrients-15-03048-f004:**
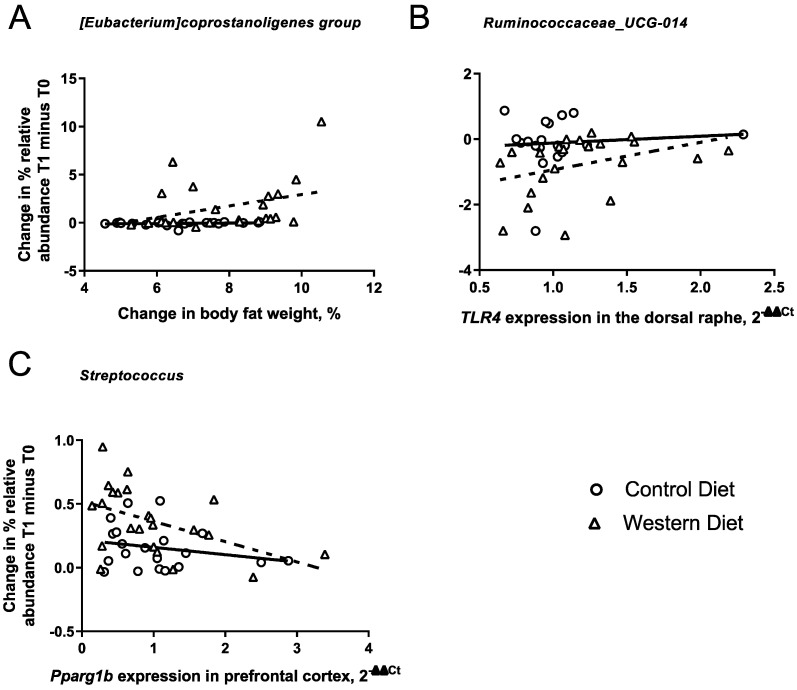
Exploratory correlations between effects of Western diet on genera abundance and physiological/gene expression changes. (**A**) Changes in genus *Coprostanoligenes* were associated with increased body fat. (**B**) Altered abundance of *Ruminococcaceae_UCG-014* was associated with lower gene expression of *Tlr4* in the dorsal raphe. (**C**) Changes in genus *Streptococcus* correlated with lower gene expression of *ppargc1b* in the prefrontal cortex. Abbreviations: Toll-Like-Receptor 4 (TLR-4), perixome-proliferator-activated-receptor-cofactor-1b (ppargc1b).

**Table 1 nutrients-15-03048-t001:** Genera for which a main effect of diet or an interaction effect between diet and genotype was observed. FDR-corrected significant values are in **bold**.

Genus	Diet		Diet × Genotype		Direction of Effect
	F	FDR p	F	FDR p	
*Atopostipes*	97.55	**6.60 × 10^−10^**	48.77	**4.40 × 10^−9^**	↑ Western in KO group
*Intestinimonas*	22.06	**0.0017**	12.39	**0.0049**	↑ Western in KO group
*Erysipelatoclostridium*	6.63	0.0771	8.94	**0.0301**	↑ Western in WT group
*Ruminiclostridium.9*	17.17	**0.0063**	1.37	0.7788	↑ Western
*Oscillibacter*	15.30	**0.0084**	2.24	0.6131	↑ Western
*Ruminococcaceae.uncultured*	9.35	**0.0367**	1.03	0.8702	↑ Western
*Ruminiclostridium.5*	9.25	**0.0367**	0.32	1	↑ Western
*Coprostanoligenes.group*	10.47	**0.0287**	1.77	0.7185	↑ Western
*Lachnospiraceae(f).A2*	11.59	**0.0240**	5.86	0.1070	↑ Western
*Peptococcaceae(f).uncultured*	9.39	**0.0367**	1.40	0.7788	↑ Western
*Streptococcus*	10.93	**0.0258**	0.30	1	↑ Western
*Staphylococcus*	9.23	**0.0367**	3.76	0.2767	↑ Western
*Alistipes*	8.43	**0.0442**	0.62	0.9626	↑ Western
*Bilophila*	12.60	**0.0179**	0.20	1	↑ Western
*Faecalibaculum*	16.32	**0.0069**	0.03	1	↑ Control
*Romboutsia*	14.11	**0.0113**	6.69	0.0888	↑ Control
*RF39(f).uncultured*	10.98	**0.0258**	4.42	0.2145	↑ Control
*Turicibacter*	21.84	**0.0017**	5.66	0.1070	↑ Control
*Dubosiella*	8.89	**0.0402**	1.12	0.8292	↑ Control
*Ruminococcaceae.UCG.014*	8.56	**0.0440**	1.94	0.6574	↑ Control

## Data Availability

Data available upon request.

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
