# Peer review of "Serotonin Transporter (SERT) Expression Modulates the Composition of the Western-Diet-Induced Microbiota in Aged Female Mice"

_nutrients, 2023, doi:10.3390/nu15133048_

Round 1
Reviewer 1 Report
This article represents that changes in gut microbiota and gene expression of some key markers of inflammatory effects of the WD in the SERT model. The manipulation of these genera may prove useful in individuals with the short SERT allele.
Line No.101-102 this sentence "as well as increased abundance of the species Turicibacter sanguinis regulating triglyceride metabolism and adipose tissue" needs a reference.
Higher figure resolution throughout the manuscript is needed. This can be performed by using a paste special function when copying figures from GraphPad prism.
Lines No.216, and 308 need to be checked for a spacing format as it is misalignment.
Author Response
Reviewer 1
This article represents that changes in gut microbiota and gene expression of some key markers of inflammatory effects of the WD in the SERT model. The manipulation of these genera may prove useful in individuals with the short SERT allele.
Reply: We wish to thank the reviewer for such constructive feedback, and we welcome the opportunity to respond. Our changes to the manuscript text are italicized below. The page references for the changes that have been made in the manuscript are to the version with track changes activated, and not to the clean manuscript.
PLEASE NOTE: the manuscript has been extensively edited by a native English speaker for clarity.
Line No.101-102 this sentence "as well as increased abundance of the species Turicibacter sanguinis regulating triglyceride metabolism and adipose tissue" needs a reference.
Reply: We have adapted this sentence, Introduction, page 5:
‘A genetic lack of SERT can also lead to decreased abundance in Bifidobacteria that regulates gut functionally important synthesis of short-chain-fatty-acids, and increased abundance in spore-forming bacteria (such as Turicibacter sanguinis), promoting serotonin production by enterochromaffin cells and regulating triglyceride metabolism and adipose tissue (Fung et al., 2019; Ye et al., 2021).’
Higher figure resolution throughout the manuscript is needed. This can be performed by using a paste special function when copying figures from GraphPad prism.
Reply: Higher resolution images have been provided.
Lines No.216, and 308 need to be checked for a spacing format as it is misalignment.
Reply: We thank the reviewer for this comment. We have checked spacing and alignment throughout the manuscript and corrected it accordingly.

Reviewer 2 Report
This was an interesting paper discussing the importance and effects of serotonin, gut microbiota, physiology and a Western diet. The information can be helpful for treating diseases (psychiatric and metabolic) and identifying specific factors in the microbiome that influence health (positively and negatively).
Notes:
Abstract
Line 26
HET (possibly define as heterozygous)
Introduction
Line 5
OECD area (what is this area?)
Line 69
“For example, germ-free mice have been shown to remain lean on a Western diet (Bäckhed et al., 2007)”
Can this be explained more, how are they germ-free, and what germs induce obesity?
Methods
Line 126-127
“Reversed 12 h light-dark cycle (lights on: 21:00 h)”
Can this be explained in more detail (lights on and off)
2.4 Bioinformatics
Line 169-178
In general, the font and spacing should be consistent throughout the paper
2.7 Compositional analyses on the taxonomic data
Line 200-226
In general, the font and spacing should be consistent throughout the paper
3.3. Compositional analyses of gut microbiota
Line 260
Define FDR
3.6. Diet * Genotype interactions
Line 300-320
In general, the font and spacing should be consistent throughout the paper
Discussion
4.1. Summary
Line 347-348
“Moreover, abundance of 11 genera increased and six decreased by the Western Diet”
Perhaps explain in more detail in general whether the increased and decreased genera were positive or negative.
4.2. Effects of Western diet on gut microbiota
Line 367-369
“Seven out of the 11 genera whose abundance was increased in the groups of mice receiving WD were previously reported to be increased in adult mice after similar interventions of housing on Western-type or cafeteria-like diets”
Can the effects on housing be explained more?
Conclusions
Line 514
TLR4 should be italicized
Can be improved, perhaps read through more carefully and add in needed changes.
Author Response
Reviewer 2
This was an interesting paper discussing the importance and effects of serotonin, gut microbiota, physiology and a Western diet. The information can be helpful for treating diseases (psychiatric and metabolic) and identifying specific factors in the microbiome that influence health (positively and negatively).
Reply: We want to thank the reviewer for their constructive feedback.
Notes:
Abstract
Line 26
HET (possibly define as heterozygous)
Reply: The abbreviation ‘HET’ is defined in the Background section of the Abstract, Page 3:
‘Aged female mice with complete (Sert-/-: KO) or partial (Sert+/-: HET) loss of SERT exhibit more pronounced negative effects following their exposure to a Western diet (WD) in comparison to wild type (Sert+/+: WT) animals.’
Following these definitions, we refer to the heterozygous group as HET throughout the manuscript, which we feels is easier to understand in the narrative. We have now reintroduced these definitions again in the main text, Introduction, page 6:
‘In a previous study, 12-month-old female mice with complete (Sert-/-: KO) or partial (Sert+/-: HET) loss of SERT were found to have overly elevated metabolic, molecular, and behavioural responses to a 3-week-long period on the WD, when compared with wild type (Sert+/+: WT) animals (Veniaminova et al., 2020).’
Introduction
Line 5
OECD area (what is this area?)
Reply: The OECD area are countries part of the “Organisation for Economic Co-operation and Development (OECD)” It is an intergovernmental organisation with 38 Member countries,[1][4] founded in 1961 to stimulate economic progress and world trade (source: Wikipedia). This reference covers statistics on overweight and obesity in these countries, equivalent of first-world countries. We have added this clarification in the Introduction, page 4:
‘Today, more than one in two adults, and nearly one in six children, are overweight or obese in the OECD area, i.e. first-world countries (Khabazkhoob et al., 2017), (Khabazkhoob et al., 2017)…”
Line 69
“For example, germ-free mice have been shown to remain lean on a Western diet (Bäckhed et al., 2007)”
Can this be explained more, how are they germ-free, and what germs induce obesity?
Reply: In addition to host influences, such as stomach acid and gut peristalsis, gut bacteria are critically involved in the digestion of nutrients and freeing nutritional compounds such as amino acids, fats, vitamins etc from the food matrix upon which they can be absorbed into the blood circulation. The experiment by Bäckhed et al., 2007 shows that having microbes in the gastrointestinal tract is necessary for the development of obesity. Germ-free mice or Gnotobiotic animals are microbiologically sterile in that no living organisms can be cultured from the gut. Strict husbandry protocols and stringent testing regimens are required to maintain and confirm the germ-free state.
We have added clarification in the Introduction, page 4:
‘For example, germ-free mice (mice born and raised in a sterile environment, preventing the colonization of the gut with microbes) have been shown to remain lean on a Western diet (Bäckhed et al., 2007).’
Methods
Line 126-127
“Reversed 12 h light-dark cycle (lights on: 21:00 h)”
Can this be explained in more detail (lights on and off)
Reply: A reversed light and dark schedule was implemented so as not to disturb the natural circadian cycle of the mice, who are nocturnal animals. Hence the lights are switched on automatically at 21:00 and off at 09:00 to accommodate this schedule. We have clarified this in the Methods section, page 7:
‘Mice were housed in groups of five under standard laboratory conditions (22 ± 1°C, 55% humidity, food and water ad libitum) and a reversed 12 h light-dark cycle (lights on: 21:00 h, lights off: 09:00 h, to accommodate their natural circadian rhythm).’
2.4 Bioinformatics
Line 169-178
In general, the font and spacing should be consistent throughout the paper
2.7 Compositional analyses on the taxonomic data
Line 200-226
In general, the font and spacing should be consistent throughout the paper
3.3. Compositional analyses of gut microbiota
Line 260
Define FDR
3.6. Diet * Genotype interactions
Line 300-320
In general, the font and spacing should be consistent throughout the paper
Reply: We have aligned the layout in terms of spacing and subheadings consistently in italic, not bold font. We now define the term FDR at the point at which it is first used in the manuscript, in the Methods section, page 11:
‘In case of FDR (False Discovery Rate)-corrected significant main or interaction effects,….’
Discussion
4.1. Summary
Line 347-348
“Moreover, abundance of 11 genera increased and six decreased by the Western Diet”
Perhaps explain in more detail in general whether the increased and decreased genera were positive or negative.
Reply: Generally, an increase in abundance in certain genera after animals are fed a Western Diet suggests that these bacteria grow better on the components of the Western Diet (increased fat and cholesterol) and a decrease indicates the numbers of bacterial genus diminish in absence of sufficient (healthy) nutrients that were previously available in the gut. Critically the changes in abundance of these genera likely contribute to the negative metabolic consequences of this unhealthy diet on the host, which we show e.g. in the correlation with increased body fat and the genus [Eubacterium]coprostanoligenes group.
We have added this interpretation to the Discussion sub-section “Summary” on page 19:
‘Moreover, the abundance of 11 genera increased and six were decreased by consumption of the Western Diet, which is likely to be a contributing factor to the negative impact of Western diet on host health. Indeed, Western-diet-induced changes in the genera Coprostanoligenes, Streptococcus and Ruminococcaceae_UCG-014 correlated with important negative physiological consequences of the dietary challenge in mice, such as an increased of a mass of visceral fat, a decrease of mitochondrial function marker Ppargc1b mRNA levels in the prefrontal cortex, and expression of Tlr4 mRNA in the dorsal raphe..’
4.2. Effects of Western diet on gut microbiota
Line 367-369
“Seven out of the 11 genera whose abundance was increased in the groups of mice receiving WD were previously reported to be increased in adult mice after similar interventions of housing on Western-type or cafeteria-like diets”
Can the effects on housing be explained more?
Reply: In this context, housing simply means the animals that received the WD (Western Diet) were housed together in the same cage, and were caged separately from those receiving the CD (Control Diet). We have changed the wording for clarification in the Discussion section, page 19:
‘Seven out of the 11 genera whose abundance was increased in the groups of mice fed the Western diet were previously reported to be increased in adult mice after their exposure to similar dietary regimens on a Western-type or cafeteria-like diet.’
Conclusions
Line 514
TLR4 should be italicized
Reply: We have adapted this. We thank to thank the reviewer for being spotting the inconsistencies in the layout.
